# Larvicide Activity on *Aedes aegypti* of Essential Oil Nanoemulsion from the *Protium heptaphyllum* Resin

**DOI:** 10.3390/molecules25225333

**Published:** 2020-11-16

**Authors:** Cleidjane Gomes Faustino, Fernando Antônio de Medeiros, Allan Kardec Ribeiro Galardo, Alex Bruno Lobato Rodrigues, Rosany Lopes Martins, Yuri de Medeiros Souza Lima, Josean Fechine Tavares, Marcos Antônio Alves de Medeiros, Jader dos Santos Cruz, Sheylla Susan Moreira da Silva de Almeida

**Affiliations:** 1Postgraduate Program of Pharmaceutical Innovation, Laboratory of Pharmacognosy and Phytochemistry, Federal University of Amapá, Highway Juscelino Kubitschek, Km 02, Macapá 68903-197, Brazil; cgfenfermagem@gmail.com; 2Postgraduate Program in Health Sciences, Federal University of Amapá, Highway Juscelino Kubitschek, Km 02, Macapá 68903-197, Brazil; fernandomedeiros1973@gmail.com (F.A.d.M.); yurimedeiros88@gmail.com (Y.d.M.S.L.); 3Institute of Scientific and Technological Research of the State of Amapá, Highway Juscelino Kubitschek, Km 10, Macapá 68903-419, Brazil; allangalardo@gmail.com; 4Postgraduate Program in Biodiversity and Biotechnology Network BIONORTE, Laboratory of Pharmacognosy and Phytochemistry, Federal University of Amapá, Highway Juscelino Kubitschek, Km 02, Macapá 68902-280, Brazil; alexrodrigues.quim@gmail.com (A.B.L.R.); rosyufpa@gmail.com (R.L.M.); 5Postgraduate Program in Natural and Synthetic Products at the Federal University of Paraíba, Cidade Universitária, João Pessoa 58051-900, Brazil; josean@ltf.ufpb.br; 6Nova Esperança (FAMENE) School of Medicine, Frei Galvão, Gramame, João Pessoa 58067-695, Brazil; marcosmaa@gmail.com; 7Faculty of Medical Sciences of Paraíba (FCM) BR 230, Km 9, Intermares-Cabedelo, Paraíba 58310-000, Brazil; 8Postgraduate Program in Biochemistry and Immunology Federal University of Minas Gerais, Avenida Antônio Coelho, Minas Gerais 31270-901, Brazil; jadercruzytrio@gmail.com

**Keywords:** colloidal system, natural compounds, nanotechnology

## Abstract

The aim of this work was to prepare a nanoemulsion containing the essential oil of *Protium heptaphyllum* resin and to evaluate the larvicidal activity and the residual larvicidal effect against *Aedes aegypti.* The essential oil was identified by gas chromatography coupled to a mass spectrometer, and the nanoemulsions were prepared using a low-energy method and characterized by photon correlation spectroscopy. The results indicated the major constituents as *p*-cimene (27.70%) and α-Pinene (22.31%). Nanoemulsions had kinetic stability and a monomodal distribution in a hydrophilic-lipophilic balance of 14 with particle diameters of 115.56 ± 1.68 nn and zeta potential of −29.63 ± 3.46 mV. The nanoemulsion showed larvicidal action with LC_50_ = 2.91 µg∙mL^−1^ and residual larvicidal effect for 72 h after application to *A. aegypti* larvae. Consequently, the nanobiotechnological product derived from the essential oil of *P. heptaphyllum* resin could be used against infectious disease vectors.

## 1. Introduction

Tropical diseases transmitted by vectors are considered a serious public health problem in developing countries. *Aedes aegypti* is a major vector responsible for transmitting arboviruses, such as Dengue, Chikungunya, and Zika, which have high mortality rates, in addition to leaving sequelae, such as persistent arthralgia in the case of Chikungunya, and microcephaly by Zika [1,2]. The use of chemical products to combat vectors in the larval phase has been one of the main tools for the control of these diseases. In this sense, the use of plants with insecticidal action is an old practice, however it has been extensively explored even in recent years [1,2].

The natural products obtained from vegetable raw materials offer a wide variety of molecules with structural diversity and biological activities, such as: herbicidal, insecticidal, fungicidal actions, among other pharmacological activities.

Among natural products for insecticidal purposes, essential oils deserve to be highlighted, either for their low toxicity to the environment, as well as to their high lipophilicity [3], which makes them an excellent alternative for this purpose. Studies have shown that several essential oils have an excellent larvicidal and repellent effects towards different species of insects [4].

With regard to environmental issues, the search for new natural pesticides has been strongly encouraged, given that these products are safer than synthetic insecticides [5]. In general, these natural pesticides are less resistant and degrade more quickly, causing less damage to the environment [5].

*P. heptaphyllum* Aubl. Marchand belongs to the family Burseraceae and is popularly known by the name of “Breu Branco”. It is found in the Amazon rainforest and exploited commercially for the commercial value of wood, or by producing an oleoresin exudate rich in volatile substances, which are used for various purposes, including as a natural insect repellent [6,7].

Phytochemical studies, combined with nanotechnology, are an important tool for the discovery of new active substances, and enhancement of therapeutic efficacy and/or biological actions that can be used in most diverse areas, including control of mosquitoes that cause diseases to humans [8].

Nanoemulsions are colloidal systems that unite the concept of natural products encapsulated in a nanotechnological system, characterized by an average diameter ranging from 20 nm to 200 nm showing kinetic stability. Its reduced size hinders deformations and decreases instability by Ostwald ripening effect [9,10,11,12].

Nanoinsecticides based on natural products, such as essential oils, have advantages that can only be achieved on a nanoscale; including physical stability, protection against chemical degradation, controlled release, solubility in water, and control loss through evaporation [12].

Thus, the objective of this study is three-fold: (1) To identify the phytochemical constituents of the essential oil obtained from the *P. heptaphyllum* resin; (2) to develop nanoemulsions containing the essential oil and to characterize the most stable formulation; and (3) to evaluate its larvicidal activity and its residual effect against *A. aegypti* larvae.

Nanoemulsions prepared from the essential oil of the *P. heptaphyllum* resin, showed kinetic stability, larvicidal activity and residual effect in *A. aegypti*. The results obtained indicated larvicidal action at a concentration of 20 µg∙mL^−1^ in 48 h and reached 100% mortality. In the residual test, concentrations of 20 µg∙mL^−1^ and 40 µg∙mL^−1^ showed mortality for three days; while doses of 60, 80, and 100 µg∙mL^−1^ exhibited mortality for 11 days.

## 2. Results

### 2.1. Chemical Composition

The extraction of essential oil from *P. heptaphyllum* resin yielded 0.69 ± 0.085 (*m*/*v*) and presented a transparent color and strong aroma characteristic of this species. The phytochemical profile revealed the presence of 20 compounds that corresponded to 100.01% of the chemical composition of the essential oil, of which *p*-cymene (27.70%) and α-pinene (22.31%) were the major constituents.

Table 1 shows all constituents identified, with their retention times (R_T_), percentage of peak area (%) and retention index (RI) from the gas chromatography coupled to mass spectrometry.

### 2.2. Nanoemulsion Stability of Essential oil from P. heptaphyllum Resin

The optimized nanoemulsion containing essential oil from *P. heptaphyllum* resin were obtained by the low-energy method, non-heating and solvent-free, and presented a bluish reflection, without sedimentation or phase separation, revealing macroscopic visual characteristics of a good nanoemulsion (Figure 1).

Table 2 indicates the distribution of droplet size, polydispersity index, and zeta potential value obtained from the nanoemulsion prepared in hydrophilic-lipophilic balance (HLB) 14.

The photon correlation spectroscopy analysis revealed that the hydrodynamic diameters of the nanoemulsion of the essential oil of the *P. heptaphyllum* resin had an average size on day zero of 109 ± 0.75 nm, polydispersity index of 0.29 ± 0.007, and zeta potencial of −21.7 ± 1.10 mV.

Droplet size stability was observed over the days. After 14-day period, it was noted that the particle size was 115.56 ± 1.68 nm, polydispersity index of 0.40 ± 0.005, and zeta potential of −29.63 ± 3.46 mV. The polydispersity index informs about homogeneity in the distribution of droplet size in the medium. Values similar to those obtained in this study show that the chances of obtaining the formation of monodispersed systems are greater.

### 2.3. Larvicidal Activity of Nanoemulsions Prepared Using Essential Oil of P. heptaphyllum Resin

Table 3 shows the mortality rate of the 3rd instar of *A. aegypti* larvae in contact with different concentrations of nanoemulsion containing the essential oil of *P. heptaphyllum* resin in 24 and 48 h of exposure. The results showed that larvicidal activity of the nanoemulsion of the essential oil from *P. heptaphyllum* resin has the potential to control *A. aegypti* mosquito, easily observed when comparing the results with the positive control Temephos which is a commercially available larvicide

The probit analysis showed a statistically significant relationship between mortality rates at 24 and 48 h of exposure, with the concentrations of nanoemulsion evaluated to a 95% confidence level. Therefore, the mortality rate is concentration-dependent. Mortality rates induced by nanoemulsion were time-dependent and showed LC_50_ in 48 h approximately 17 times smaller than LC_50_ in 24 h of exposure and demonstrate a cumulative effect of the essential oil Table 3. When compared LC_50_ after 48 h exposure to nanoemulsion of the essential oil of *P. heptaphyllum* resin to Temephos LC_50_, it was 51 times more potent, demonstrating a high lethal effect of the nanoemulsion against *A. aegypti* mosquito larvae.

### 2.4. Residual Effect Results

Figure 2 shows the percentage of mortality (%) 3rd instar *A.*
*aegypti* larvae_,_ observed for a period of 11 days (264 h) after application of nanoemulsion of the essential oil of *P. heptaphyllum* resin.

In the 48 h interval, the nanoemulsion at 20 µg∙mL^−1^ showed 20% mortality. The larvicidal effects of this concentration decreased by 78% when compared to the larval mortality in 24 h of testing. The concentration of 20 µg∙mL^−1^ did not cause mortality within 96 h of preparing the solution, and this result indicates that this concentration did not release essential oil from the nanoemulsion from that moment on.

## 3. Discussion

### 3.1. Chemical Composition and Yield of Essential Oil of P. heptaphyllum Resin

The percentage yield of essential oil of *P. heptaphyllum* resin obtained in this work is in accordance with the percentage described in the scientific literature, whose values do not exceed 2% of yield for species of the genus *Protium* [14].

The Burseraceae family has predominantly monoterpenes and sesquiterpenes in its chemical composition. The main monoterpenes found in resin essential oils are derived from the *p*-menthane (monocyclic) and pinane (bicyclic) skeletons. In relation to sesquiterpenes, the most identified in Burseraceae are derived from skeletons of muurulan, selinane, humulan and caryophyllan [15].

Research carried out in the state of Amazonas (Brazil), detected a predominance of monoterpenes *p*-cymene and terpinolene in the essential oil of *P. heptaphyllum* resin [16,17]. In specimens of *P. heptaphyllum* from the state of Ceará (Brazil) were identified *α*-felandrene, limonene, α-pinene, and terpinolene as the major components [1].

The scientific literature indicates *p*-cymene and terpinolene as major components of the essential oil of two subspecies of *P. heptaphyllum* (Aubl.) Marchand (*Protium heptaphyllum* subsp. *cordatum* (Huber) D.C. Daly; and *Protium heptaphyllum* subsp. *ulei* (Swart) Daly), which presented a yield of 39.93% for *p*-cymene, and 42.31% for terpinolene. The results found in this study differed in the content of *p*-cymene (27.70%) and in the presence of *α*-pinene (22.31%) as a major constituent [18].

In another study using the essential oil of the resin it was indicated in fresh plants the present of terpinolene in the concentration of 28.2%–69.7% [2]. In the matured resin it was found *p*-cymene ranging from 18.7% to 43.0%, and *p*-cymen-8-ol ranging from 8.2%–31.8% [2].

The difference in chemical constituents may be the result of environmental factors that may influence the composition of secondary metabolites in plants, for instance: seasonality, collection time, rainfall index, ultraviolet radiation, altitude, average temperature, soil nutrients, and defenses against herbivores. Even similar environmental conditions between micro-regions, small variations can influence the chemical diversity of the species and justifies the differences in chemical constituents.

### 3.2. Stability of the Essential Oil Nanoemulsion of the P. heptaphyllum Resin

The titration method used in this study is associated with a phase inversion composition process to produce nanostructures. The low-energy emulsification methods are preferable to obtain stable nanoemulsions for use in the delivery of secondary metabolites that are sensitive to temperature rise, such as essential oils, with monomodal particle size [19].

Ostwald ripening is the main mechanism involved in destabilizing nanoemulsions, this phenomenon occurs when some component of the dispersed oily phase has some degree of solubility in the external phase and then smaller particles fuse and form structures with larger hydrodynamic sizes [12]. However, the results obtained suggest that were maintained the particle size, polydispersity index and zeta potential, and indicates the kinetic stability and protection against Ostwald ripening for the essential oil nanoemulsion of the *P. heptaphyllum* resin during a period of 14 days.

Transparent or translucent emulsions with a particle size distribution between 20–200 nm and a zeta potential around ±30 mV as described in this study are normally considered stable [20]. Another characteristic observed in the nanoemulsions produced in this work was the Tyndall effect. This bluish reflection is the result of the light scattering and indicates the formation of a kinetic system with nanoscale particles that are suspended in an aqueous medium [21,22].

Analyzing the kinetic stability of the nanoemulsion it is observed that the nanoemulsion did not present evidence of instability. Thus, it is possible to conclude that the essential oil nanoemulsion from the *P. heptaphyllum* resin could be applied in the chemical control of *A. aegypti* larvae.

### 3.3. Larvicidal Evaluation of P. heptaphyllum Nanoemulsion in A. aegypti Larvae

The larvicidal effects of the nanoemulsion of *P. heptaphyllum* resin against *A. aegypti* may be associated with the presence of a chemical constituent or through the synergistic action of different secondary metabolites present in the essential oil [23].

The literature indicates that the larvicidal activity of terpenes such as α-pinene, β-pinene, linalool, and eugenol is related to the structural characteristics of molecules, such as the exocyclic double bond present in β-pinene [24]. In the chemical composition of the essential oil of the *P. heptapyllum* resin was found α-pinene with 22.31% content which suggests the possibility that α-pinene is involved in the larvicidal action of nanoformulation. The γ-muurolene sesquiterpene is another minor chemical constituent present in essential oil that has an exocyclic double bond that may be involved with the larvicidal action. Thus, it can be suggested that the larvicidal action found in the nanoemulsion of the essential oil of *P. heptapyllum* resin may be related both to the presence of chemical constituents with recognized insecticidal action, such as *α*-pinene and γ-muurulene, or by the synergistic action of all constituents [24].

The larvicidal action mechanisms of essential oils on insects may be related to the action on digestive and/or neurological enzymes, in addition to possible interactions in the integumentary system [25]. Emphasizing the importance of the relationship between the chemical structure and biological activity of the compounds, it is possible to conclude that the increase in the lipophilicity of the chemical constituent influences the penetration in the integumentary system [26]. In this context, the main constituents identified have a high lipophilicity and corroborate the larvicidal action found in this study.

There are evidences that nanoemulsions based on essential oil had the potential in terms of mosquito control, as can be seen in the studies described below.

There is evidence that nanoemulsions based on essential oil had the potential in terms of mosquito control. The essential oil nanoemulsion of the *Rosmarinus officinalis* caused mortality of 80 ± 10% after 24 h of exposure and 90 ± 10% after 48 h of exposure at 250 µg∙mL^−1^ concentration [27].

While nanoemulsion containing *Pterodon emarginatus* essential oil showed a larvicidal potential against *A. aegypti* with mortality of 66.7 ± 12.5% in 24 h at the highest tested concentration (500 μg∙mL^−1^) [28].

Study of the larvicidal activity of *Carlina acaulis* essential oil and the nanoemulsion prepared from it, against the larvae of *Lobesia botrana*, showed LC_50_ = 7.29 (6.90–7.60) and 9.04 (8.60–9.70), respectively, and LC_90_ = 10.92 (9.70–13.60) and 17.70 (15.40–27.50), respectively [29], showing a lower efficiency for nanoemulsion when compared to essential oil *in natura*. The study of larvicidal activity, against *Culex quinquefasciatus* larvae, with the essential oil obtained from the leaves of *Smyrnium olusatrum*, isolated isofuranediene and microemulsion 750 of isofuranediene, showed LC_50_ of 18.60 (15.70–21.30); 29.20 (24.80–34.10) and 17.70 (16.40–19.20), respectively, and LC_90_ of 32.30 (29.90–35.80); 86.70 (70.80–103.10) and, 39.10 (33.10–49.70), respectively [30], demonstrating better responses of LC_50_ for the microemulsion and LC_90_ for isolated isofuranediene.

Comparing the previous results using nanoemulsions prepared from essential oils or isolated substances with the results obtained using nanoemulsion from *P. heptapyllum* resin, against *A. aegypti* larvae, it is observed that this was much more effective, reaching an LC_50_ of 48 h up to 53 times less than the best previous results.

Thus, it is possible to conclude that the nanoemulsion of *P. heptaphyllum* resin, which was prepared by the low-energy method, is a nanobiotechnological product with high larvicidal potential towards *A*. *aegypti* larvae.

### 3.4. Residual Larvicidal Effect of P. heptaphyllum Nanoemulsion

Nanoemulsion represents an advantageous biotechnological strategy in chemical control *A. aegypti* that allows the prolonged storage of volatile oils, the increased availability in aqueous medium, and the controlled release of essential oil over time, these features enables its application as a larvicidal agent with residual action in the first stage of development of mosquito larvae [31].

Studies on the residual larvicidal activity of nanoemulsions are incipient. In one of the few studies evaluating the residual larvicidal activity of nanoemulsion containing *Carapa guianensis* essential oil was observed increased mortality of *A. aegypti* larvae as a function of time in the concentration of 250 µg∙mL^−1^ [31]. In another study, the residual larvicidal activity of the whole essential oil from *C. guianensis* was maintained up to the 12 day at the concentration of 1400 µg∙mL^−1^ [32].

Our results, as far as we know, are the first to demonstrate the residual larvicidal effect on *A. aegypti* of the *P. heptaphyllum* nanoemulsion with mortality rates at concentrations lower than those previously mentioned (60, 80, and 100 µg∙mL^−1^) in 72 h exposure. The prolonged larvicidal effect is the result of the controlled release of essential oil constituents contained in the nanoemulsion over time. Thus, *P. heptaphyllum* nanoemulsion is a larvicidal agent able to stopping the development of insects in immature stages.

Some factors can interfere with the efficiency of essential oils, among which the temperature stands out. Possibly the decrease in residual response of the nanoemulsion, after 72 h, may be related to the volatilization of the essential oil constituents as a function of the temperature at which the test was performed (25–27 °C). According to a study carried out by [33], the temperature variation directly influenced LC_50_ of the *p*-cymene from 79.0 (67.0–93.0) at 20 °C to 125.0 (97.0–118.0) at 25 °C and LC_90_ 138.0 (125.0–173.0) at 20 °C to 217 (174.0–238.0) at 25 °C, against *Spodoptera littoralis* Boisd larvae. Considering that *p*-cymene is the major constituent of the essential oil of *P. hetaphyllum,* this decrease in the response can be related to its presence.

Natural insecticides are composed of plant extracts that generally contain complex mixtures of various active ingredients with different mechanisms of action [4]. Essential oils could be an alternative for the purpose of producing natural insecticides, mainly because they show low toxicity to man and the environment [3].

Therefore, the results presented here suggest that the essential oil nanoemulsion from *P. heptaphyllum* resin has prolonged larvicidal effect and could be used in the chemical control of *A. aegypti* larvae.

## 4. Materials and Methods

### 4.1. Resin Collection and Botanical Identification of P. heptaphyllum

After authorization for access to the Brazilian genetic heritage, the plant material was collected in an environmental area of preserved native forest located in the municipality of Porto Grande (State of Amapá, Brazil) under the coordinates Lat. 0°4′1″ N e Longitude 51°3′10″ W. The plant material was identified at the Herbarium Amapaense (HAMAB) of the Institute of Scientific Research and Technology of the State of Amapá (IEPA) and received the code 019059.

### 4.2. Extraction of Essential Oil from P. heptaphyllum Resin

The collected resin was washed under running water to remove the dirt. A sample of 100 g of *P. heptaphyllum* resin was homogenized in powder using porcelain mortar and hydro distilled for a period of 3 h in a Clevenger (São Paulo, SP, Brazil) apparatus to extract the essential oil through the steam distillation technique. The extraction was carried out in quintuplicate and the oil was stored in amber glass at 5 °C.

### 4.3. Gas Chromatography-Mass Spectroscopy (GC-MS) Analysis

The phytochemical profile of the essential oil of *P. heptaphyllum* resin was identified by GC-MS using Shimadzu equipment, model CGMS-QP2010 Ultra (Kyoto, Japan), equipped with an RTX-5MS capillary column (30 m × 0.25 mm, film thickness 0.25 μm), with the stationary phase being 5% diphenyl −95% dimethyl-polysiloxane. The oven temperature was programmed at 60–250 °C at a heating rate of 3 °C/min. The ion source was adjusted to 200 °C and the electronic ionization to 0.84 kV. Helium was the carrier gas at a flow rate of 1.0 mL∙min^−1^ and an inlet pressure of 57.0 KPa.

A sample of the essential oil was diluted in hexane and 10 µL of this solution was injected into the GC-MS. The relative concentrations (%) corresponding to the essential oil components were calculated using Shimadzu software. Peak identification was performed by comparing retention indices calculated from an n-alkane series (C_9_ to C_17_) and by the fragmentation pattern of the mass spectrum and compared with data from the equipment library.

### 4.4. Preparation and Characterization of Nanoemulsion

The nanoemulsion containing the essential oil of *P. heptaphyllum* resin was developed by a titration method without heating [34]. The nanoemulsion was prepared by an organic phase (75 mg surfactant + 75 mg essential oil) in an aqueous phase (2850 mg of water), with constant stirring at room temperature. The organic phase was formed by adding the surfactant to the oil and homogenized for 30 min at 750 rpm in a vortex machine (model AP59-Phoenix, Araraquara, SP, Brazil).

The surfactant was composed of a blend of sorbitan monooleato and Polysorbate 80 to produce the HLB 10, 11, 12, 13, 14, and 14.5; Polysorbate 80 to produce HLB of 15; and Polysorbate 80 and Polysorbate 20 to produce the HLB 15.5 and 16.

Then, the aqueous phase was added drop wise and the system was stirred for another 30 min. At the end, the prepared nanoemulsion presented an essential oil concentration equal to 25,000 μg∙mL^−1^ and SOR = 1:1.

The physical-chemical characterization of the nanoemulsions in each HLB produced was done through the particle size, polydispersity index and potential Zetasizer ZS equipment (Malvern, WORSTS, UK). Each nanoemulsion was diluted in deionized water (1:25 *v*/*v*) for analysis. The measurements were performed in triplicate at intervals of seven and 14 days after their production and the results were expressed in arithmetic mean ± standard deviation.

### 4.5. Evaluation of the Larvicidal Activity of the Nanoemulsion

The third instar *A. aegypti* larvae were obtained from Medical Entomology Laboratory of the Institute of Scientific and Technological Research of the State of Amapá and were maintained under standardized climatic conditions in a room measuring 12 m^2^ (3 m × 4 m), with controlled temperature and humidity (26 ± 28 °C and 80 ± 5%, respectively) and a 12 h photoperiod as recommended by the World Health Organization (WHO) [35].

The most stable nanoemulsion (HLB = 14) was used to perform the larvicidal activity. The nanoemulsion prepared at a concentration of 25,000 µg∙mL^−1^ was diluted in concentrations of 2, 5, 10, 15, and 20 µg∙mL^−1^ with a final volume of 100 mL of solution to receive 10 larvae in each beaker.

A negative control containing only the blend of nanoemulsion surfactants in HLB 14 was used in the assay and the test was performed in quintuplicate for each concentration evaluated and as a positive control, we took published LC_50_ values of the synthetic larvicide (Temephos) to compare [13]. The mortality rates were analyzed after 24 and 48 h of exposure. Larvae with total absence of voluntary movements or by exhaustive external stimulation in the beakers were considered dead.

### 4.6. Evaluation of Residual Larvicidal Activity of the Nanoemulsion

Residual effect is the ability to maintain lethal insecticidal dosages for *A. aegypti* larvae for a period of time and is measured by the incidence of daily larval mortality [36]. The most stable nanoemulsion (HLB 14) was diluted in concentrations of 20, 40, 60, 80, and 100 µg∙mL^−1^ in 100 mL solution. Ten *A. aegypti* larvae were added at each concentration and the percentage mortality was counted 24 h later. In this experiment, a blend of surfactants was used as a negative control and the test was performed in quintuplicate.

All live or dead larvae were removed from the containers by filtration after 24 h and new healthy larvae were added to the beakers to assess mortality 48 h after preparation of the solution. This process was repeated until there was no mortality in the nanoemulsion solutions.

### 4.7. Statistical Analysis

The experimental design used in the larvicidal assay and the residual larvicidal effect was completely randomized and employed five treatments with five repetitions. The data obtained in the index and in the mortality percentage were analysed by the program Prisma 5.03 (San Diego, CA, USA) in Probit analysis to determine the lethal concentration that causes mortality of 50% and 90% of the population. The probabilistic level of error employed was 95%.

## 5. Conclusions

The essential oil of *P. heptaphyllum* resin is known to have several biological activities and can provide innovative products for different industrial segments; however, no study has been carried out with the nanoemulsion of the essential oil of *P. heptaphyllum* resin.

This study demonstrated the presence of *p*-cymeno (27.70%) and α-pineno (22.31%) as major constituents. We also described the preparation of nanoemulsion by the low energy method and with kinetic stability in HLB 14 for the essential oil of *P. heptaphyllum* resin.

Larvicidal trial demonstrated that the nanoemulsion presented LC_50_ = 2.91 µg∙mL^−1^ in 24 h exposure and could be classified as highly active against *A. aegypti*. The residual larvicide test demonstrated effective controlled release of the essential oil from the nanoemulsion in 72 h and indicates the potential larvicide effect with prolonged life time of the nanoemulsion in aqueous medium.

The use of the essential oil of *P. heptaphyllum* resin deserves to be highlighted, since this nanoemulsion is based on a natural product that can be obtained by the sustainable management of biodiversity. Therefore, contributing to the conservation of this species and stimulating the development of innovative products.

## Figures and Tables

**Figure 1 molecules-25-05333-f001:**
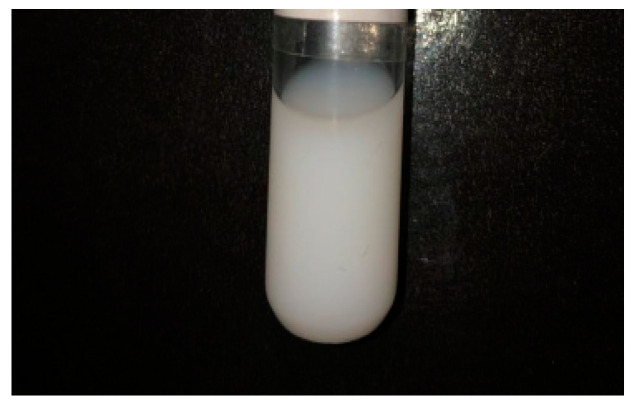
Essential oil nanoemulsion of *P. heptaphyllum* resin in hydrophilic-lipophilic balance 14 on day 0.

**Figure 2 molecules-25-05333-f002:**
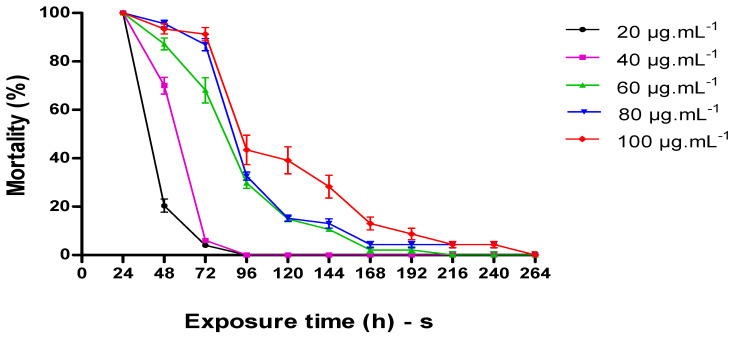
Percentage of mortality of *A. aegypti* larvae in contact with essential oil nanoemulsion of the *P. heptaphyllum* resin in the concentrations of 20, 40, 60, 80, and 100 µg∙mL^−1^ over time (264 h).

**Table 1 molecules-25-05333-t001:** Chemical constituents, retention time (R_T_), percentage of peak area, and retention index (RI) of the essential oil of *P. heptaphyllum* resin.

Peak	Compound	Retention Time	Percentage of Peak Area	Retention Index
1	2,4-Dimethylhept-1-ene	4.498	0.77	-
2	Bicyclo [3.1.0] hex-2-ene	7.101	0.95	902
3	*α*-Pinene	7.382	22.31	948
4	Camphene	7.943	1.79	943
5	Bicyclo [3.1.1] heptane	9.025	4.43	943
6	Cyclohexene	9.233	1.03	959
7	Bicyclo [4.1.0] heptane	9.967	5.92	937
8	Alpha Phellandrene	10.243	6.76	969
9	(+)-4-Carene	10.695	2.67	919
10	*p*-Cymene 3,7,7-	11.106	27.70	1042
11	Bicyclo Heptane	11.225	8.27	937
12	*β*-Phellandrene	11.300	1.82	902
13	Eucalyptol	11.348	3.44	1059
14	2-Carene	13.718	0.68	948
15	(+)-2-Bornanone	16.490	1.97	1121
16	*cis*-Dihydro-α-terpineol	16.734	3.93	1132
17	Cyclosativene	26.486	2.18	1125
18	Caryophyllene	28.602	0.62	1494
19	Δ-cadinene	32.116	1.28	-
20	*γ*-Muurolene	32.513	1.49	1435
			100.01	

**Table 2 molecules-25-05333-t002:** Droplet size, polydispersity index, and zeta potential of the nanoemulsion prepared with the essential oil of *P. heptaphyllum* resin (HBL = 14) on different days of analysis.

Day	Particle Size (nm)	Polydispersity Index	Zeta Potential (mV)
0	109.7 ± 0.75	0.29 ± 0.007	−21.7 ± 1.10
7	109.93 ± 0.97	0.28 ± 0.003	−34.66 ± 3.15
14	115.56 ± 1.68	0.40 ± 0.005	−29.63 ± 3.46

**Table 3 molecules-25-05333-t003:** Lethal concentration (LC) and its lower and upper limits of the nanoemulsion essential oil (EO) and temephos formulation for 50% and 90% mortality of *A. aegypti* larvae with 95% confidence level.

		24 h	48 h	*p*-Value
Nanoemulsion EO	LC_50_ (µg∙mL^−1^)	2.91 (0.55–4.52)	0.17 (−3.51–2.13)	<0.001
Nanoemulsion EO	LC_90_ (µg∙mL^−1^)	12.44 (10.62–15.30)	8.87 (7.23–11. 59)	<0.001
* Formulate Temephos	LC_50_ (µg∙mL^−1^)		8.70 (7.00–10.20)	

* Temephos formulation [13].

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
