# Peer review of "Larvicide Activity on Aedes aegypti of Essential Oil Nanoemulsion from the Protium heptaphyllum Resin"

_molecules, 2020, doi:10.3390/molecules25225333_

Round 1
Reviewer 1 Report
The manuscript describes the mosquito larvicidal activity of essential oil nanoemulsion from Protium heptaphyllum using Aedes aegypti.
This study is interesting because Oliverio et al. (2010) reported that the extract from P. heptaphyllum showed no to low toxicity against fourth instars of Ae. aegypti after 48 hours of exposure (Parasitol. Res. 107:403-407). Therefore, this reviewer believes that authors should include positive control such as Temephos so that readers can see how much the essential oil from P. heptaphyllum is effective compare to the commercial chemical larvicide.
Tables 3 and 4 show the same data in a different way. This reviewer suggests delete Table 3. In a same way, Figure 2 is not quite necessary.
Author Response
MACAPÁ-AMAPÁ-AMAZON, november 7, 2020.
Dear Editor and Reviewers
Initially we would like to thank the reviewers for the relevant contributions suggested in this work, we consent with those and present the necessary changes (attached letter), as suggested, we are sure that the adjustments will facilitate a better understanding of the results. The suggested manuscript entitled " Larvicide Activity on Aedes aegypti of Essential Oil Nanoemulsion from the Protium heptaphyllum Resin “, template are highlighted in red, due to facilitate their identification in the text.
We would like to apologize for not responding to emails sent, we justify not responding it due to a total blackout in the electric power system in the state of Amapá-Brazil in virtue a strong storm that occurred on Monday (02 / 11/2020) leaving 14 cities, of the 16 in the State, without electricity, internet, telephony, water and other services that depended on it, being only partially reestablished today, 07/11/2020, thus becoming the main authors and incoming correspondence during that period of time.
Thus, as the first answer sent, the authors has been working in the suggested alteration of the reviwers, in the base of the initial proposal. Even though, if there is a better comprehension for the part of this boss editor whose the paper fits best in the form of comunication, these authors do not have contradition of the proposal.
Since now, we would like to thank the comprehesion and we put ours-self in the disposition for any necessities and/or elucidation.
graciously,
Response to Reviewer 1 Comments
Point 1: Does the introdutivo provide suficiente background and include all relevant references?
Response 1: Yes
Point 2: Is the research design appropriate?
Response 2: Can be improved. The draw of the experimental search was improved in the following spots: metodology, resulte, discution e conclution. Due to attend all of the proposal objetives in a undarstanble way.
Point 3: Are the methods adequately described?
Response 3: Can be improved. In the topic 4.2 where it describes the procediment of essential oil extraction alteration were made, presented in bettwen the 279 ,282 and 283 lines. Next , in the topic Evaluation of the Larvicidal Activity Of Nanoemulsion was insered in the 323 and 324 lines, the obtained resalt with the control positive temephos, introduced on paper of Oliveira et al. (2010), as suggested by Revisor 1 that the readers can notice how the essencial oil of P. heptaphyllum is effective in conparattion with the comercial chemical larvicide.
Point 4: Are the results clearly presented?
Response 4: As suggested by reviewers the table 3 that where present the average result and the standard desviation was concluded. after alteration in the lines 130 to 132 was inserted a new table 3 referring to Lethal concentration (LC) including control positive temephos of study done by Oliveira et al. (2010). Considering that the second (2nd) figure was the only one that present the result of the assay larvicida residual the authors suggested the maintenance of those, since it was not a requirement by the reviewer. However, if he think it is better delete, he can do that without any contradiction.
Point 5: Are the conclusions supported by the results?
Response 5: In conclusion it was included a paragraph in the lines 355 to 358, enphasizing the use of natural product of the essential oil and their contributions for the enviroment and the conservation of the species.
These corrections has been translated into English with appropriate language, considering native English speakers' grammar, punctuation, spelling, and form.
Looking forward to your reply. Thank you
______________________________________________________________________
Sheylla Susan Moreira da Silva de Almeida
Reviewer 2 Report
The authors studied nanoemulsions prepared from the essential oil of the Protium heptaphyllum resin, showed kinetic stability, larvicidal activity and residual effect in Aedes aegypti. It is a relatively well-prepared manuscript that is suitable for publication in Molecules, but only after subsequent modifications.
- Results. Table 1 - add the Percentage of Peak Area below the table to make it clear what % of substances were analyzed.
- Table 3 - is confusing. Report results as average mortality.
- The discussion should be improved. It would be useful to better discuss the results with other authors (see e.g. DOI: 10.3390 / nano10091867; DOI: 10.1007 / s10340-018-01076-3) who have tested the larvicidal efficacy of EOs nanoemulsions. The effectiveness of EOs can be affected by many factors, please discuss (see e.g. DOI: 10.1016 / j.indcrop.2018.01.021). The environmental and health safety of botanical larvicides based on EOs should also be discussed.
Author Response
MACAPÁ-AMAPÁ-AMAZON, november 7, 2020.
Dear Editor and Reviewers,
Initially we would like to thank the reviewers for the relevant contributions suggested in this work, we consent with those and present the necessary changes (attached letter), as suggested, we are sure that the adjustments will facilitate a better understanding of the results. The suggested manuscript entitled " Larvicide Activity on Aedes aegypti of Essential Oil Nanoemulsion from the Protium heptaphyllum Resin “, template are highlighted in red, due to facilitate their identification in the text.
We would like to apologize for not responding to emails sent, we justify not responding it due to a total blackout in the electric power system in the state of Amapá-Brazil in virtue a strong storm that occurred on Monday (02 / 11/2020) leaving 14 cities, of the 16 in the State, without electricity, internet, telephony, water and other services that depended on it, being only partially reestablished today, 07/11/2020, thus becoming the main authors and incoming correspondence during that period of time.
Thus, as the first answer sent, the authors has been working in the suggested alteration of the reviwers, in the base of the initial proposal. Even though, if there is a better comprehension for the part of this boss editor whose the paper fits best in the form of comunication, these authors do not have contradition of the proposal.
Since now, we would like to thank the comprehesion and we put ours-self in the disposition for any necessities and/or elucidation.
graciously,
Response to Reviewer 2 Comments
Point 1: Does the introduction provide suficiente background and include all relevant references?
Response 1: Can be improved. The introduction was improved by insection of information about essentiol oil and its use as an larvicide agent and repellent against the difference among the kind of insects, shown in the lines 48 to 51.
Point 2: Is the research design appropriate?
Response 2: Yes
Point 3: Are the methods adequately described?
Response 3: Yes
Point 4: Are the results clearly presented?
Response 4: Can be improved. In table 1, presented in the template lines 87 e 88, was inserted the percentage of the total área of peaks 100%, as the schedule original.
Related to results of the table 3, regarding the average of standart desviation, as questioned by Reviewer 2 that they would be confused, was complete view the suggestion by reviewer 1 that sugested this exclusion. Thus, is being presented a new Table 3 - Lethal concentration (LC) and its lower and upper limits of the nanoemulsion essential oil (EO) and formulate temephos for 50% and 90% mortality of A. aegypti larvae with 95% confidence level. Included in 130 to 132.
The discution were improved from the insert data of the article doi: 10.3390 / nano10091867; doi: 10.1007 / s10340-018-01076-3), sugested by reviewer 2, presented in the lines 214 e 215, that referred to studies based in nanoemulsion prepared with the essential oil, focusing in Study of the larvicidal activity of Carlina acaulis essential oil and the nanoemulsion prepared from it, against the larvae of Lobesia botrana, when compared to essential oil in natura. Also it was discussed the larvicide activity, against larvae of quinquefasciatus, with essential oil obtained form leaves of Smyrnium olusatrum, isofuranediene isolated in microemultion 750 of isofuranediene, as suggested doi: 10.1016 / j.indcrop.2018.01.021 located in the lines 223 a 231. Was increased in discution how much the nanoemultion of the essential oil of Protium heptaphyllum resin is effective compared to the results shown in reference above in the lines 232 to 235.
As regards to the factors that interfere in the effectiveness of the essential oil, as questioned by the reviewer 2, it was pesent in the item 3.4 regarding the assay residual larvicide in the lines 256 a 263, enphasizing the temperature.
When the suggestion of the reviewer 2 refering the inclusion of the environmental safty and the human health based on essential oil for the environmental and the human being presente in the line 264 to 267.
Point 5: Are the conclusions supported by the results?
Response 5: Can be improved. In conclusion it was included a paragraph in the lines 355 to 358, enphasizing the use of natural product of the essential oil and their contributions for the enviroment and the conservation of the species.
These corrections has been translated into English with appropriate language, considering native English speakers' grammar, punctuation, spelling, and form.
Looking forward to your reply. Thank you
______________________________________________________________________
Sheylla Susan Moreira da Silva de Almeida
Round 2
Reviewer 1 Report
The revised manuscript is acceptable for publication.